# Quantum Physics Literacy Aimed at K12 and the General Public †

**Caterina Foti** [1,2,*] , **Daria Anttila** [3] , **Sabrina Maniscalco** [1,2,4] **and Maria Luisa Chiofalo** [5,*]

1   QTF Centre of Excellence, Department of Applied Physics, School of Science, Aalto University, FI-00076 Aalto, Finland; sabrina.maniscalco@helsinki.fi
2   Algorithmiq Oy, Linnankatu 55 K329, 20500 Turku, Finland
3   QTF Centre of Excellence, Turku Centre for Quantum Physics, Department of Physics and Astronomy, University of Turku, FI-20014 Turku, Finland; daria.alekseeva@utu.fi
4   QTF Centre of Excellence, Department of Physics, Faculty of Science, University of Helsinki, FI-00014 Helsinki, Finland
5   Dipartimento di Fisica Enrico Fermi, Università di Pisa and INFN, Largo B. Pontecorvo 3, I-56127 Pisa, Italy
*   Correspondence: caterina.foti@aalto.fi (C.F.); maria.luisa.chiofalo@unipi.it (M.L.C.)
†   This paper is an extended version from the proceeding paper: Caterina Foti, Daria Anttila, Sabrina Maniscalco and Maria Luisa Chiofalo. Quantum Physics Literacy Aimed at K12 and General Public. In Proceedings of the 1st Electronic Conference on Universe, online, 22–28 February 2021.

**Abstract:** Educating K12 students and general public in quantum physics represents an evitable must no longer since quantum technologies are going to revolutionize our lives. Quantum literacy is a formidable challenge and an extraordinary opportunity for a massive cultural uplift, where citizens learn how to engender creativity and practice a new way of thinking, essential for smart community building. Scientific thinking hinges on analyzing facts and creating understanding, and it is then formulated with the dense mathematical language for later fact checking. Within classical physics, learners' intuition may in principle be educated via classroom demonstrations of everyday-life phenomena. Their understanding can even be framed with the mathematics suited to their instruction degree. For quantum physics, on the contrary, we have no experience of quantum phenomena and the required mathematics is beyond non-expert reach. Therefore, educating intuition needs imagination. Without rooting to experiments and some degree of formal framing, educators face the risk to provide only evanescent tales, often misled, while resorting to familiar analogies. Here, we report on the realization of QPlayLearn, an online platform conceived to explicitly address challenges and opportunities of massive quantum literacy. QPlayLearn's mission is to provide multilevel education on quantum science and technologies to anyone, regardless of age and background. To this aim, innovative interactive tools enhance the learning process effectiveness, fun, and accessibility, while remaining grounded on scientific correctness. Examples are games for basic quantum physics teaching, on-purpose designed animations, and easy-to-understand explanations on terminology and concepts by global experts. As a strategy for massive cultural change, QPlayLearn offers diversified content for different target groups, from primary school all the way to university physics students. It is addressed also to companies wishing to understand the potential of the emergent quantum industry, journalists, and policymakers needing to seize what quantum technologies are about, as well as all quantum science enthusiasts.

**Keywords:** quantum physics education; methods for physics teaching; games with a purpose

## 1. Introduction

Developed to describe atomic-scale phenomena, quantum mechanics is the most fundamental of all physical theories, and it has never been disproved so far. It describes physical processes, from the cosmology of the early universe to how transistors and lasers work, but understanding it is a journey into the rabbit hole, through the weird and

bizarre. Around 100 years ago, some of the world's greatest scientists, in their study of the microscopic building blocks of matter, started to uncover a new world that defies common sense and logic. Since then, our understanding of Nature has dramatically changed, and we have been forced to start thinking differently and questioning basic concepts taken for granted. Indeed, even within classical physics, abstract approaches to teaching and learning processes still remain most practiced, and a vast education research literature is devoted to conceive and assess the effectiveness of alternative frameworks, centered on the idea that learners can be actively engaged and their intuition educated via classroom demonstrations of everyday-life phenomena [1–5], with the advantage of structurally hinging the learning process on the observation as a pillar in physics thinking. In quantum physics, on the other hand, our intuition can be too often educated only by means of advanced mathematical forms, with not even classroom demonstrations easily at reach. Thus, quantum mechanics as a theory necessarily calls for a different way of practicing scientific thinking in educational, popularization, and research contexts. With the words of Nobel laureate Tony Leggett, quantum mechanics is much more than just a theory: it is a new way of looking at the world, it forces us to move out of our comfort zone, going beyond conventional frames of mind, and opening up to new things. It, therefore, represents a valuable opportunity to search for new learning frameworks in formal education, and to enhance citizens' critical thinking in popularization. Moreover, we are currently in the midst of the so called "Second Quantum Revolution": the development of quantum mechanics has led to quantum technologies which are now entering the market. They hold the promise to affect dramatically our life overturning everything, from drug development, to cryptography, to data science and Artificial Intelligence (AI). This is the reason why it becomes crucial to develop methods to make quantum physics and technology understandable for society. Above all, understanding technology makes us aware of our capabilities and limitations. In fact, knowledge is an essential tool for the citizens to be able to operate choices, that is a form of power and freedom for individuals, and of smartness for communities.

Although it represents an exciting opportunity, quantum literacy for K12 students and a general public is particularly challenging, since the methodology exploited for learning quantum science usually relies upon traditional approaches, grounded on solid mathematical bases. Therefore, even restricting to high-school level, it is not easy to involve students, since many of them are scared by mathematics, and often even the most enthusiast simply do not have the proper background to delve into the quantum world. Actually, the same issues arise considering a more general public, so that demystifying quantum physics for the non-experts is still an open problem and the best strategy to achieve the goal needs still to be found.

In this paper, we present an innovative and original approach for making quantum physics accessible to the general public. The idea is inspired by the theory of multiple intelligences proposed by Howard Gardner in the 1980s [6], in particular, fostering the concept that each person possesses a number of different intelligences, that are competences acted in relationship with a context, therefore, a potentially under continuous development. As a result, efficient and effective learning and teaching processes should be tailored on these many diverse talents. These multiple intelligences work through diverse communication channels, which contemplate the use of different languages, based on words, numbers, pictures, and music, but also on the importance of social interactions, physical motion, introspection. The tool that we developed is QPlayLearn, an online platform containing multimedia resources for learning about quantum science and technologies in a playful way, without giving away scientific correctness. The platform is freely available for everyone (qplaylearn.com, accessed on 28 March 2021), and it allows us to create educational paths specifically tailored to the different backgrounds of the users. The first instrument developed within QPlayLearn is Quest, a free online dictionary currently containing ten quantum terms, namely key-concepts related to quantum science, and constantly expanding by the addition of new entries. In each entry, concepts are illustrated by means of

video games, animations, videos of researchers from all over the world, explanations, and illustrations of experiments, as well as some mathematical formalism, for those interested in learning the math behind quantum physics. Following the idea of the multiple intelligences, the platform offers to its explorers the freedom of choosing the learning approach that better suits to their personal preferences, via three different options: Play (intuitive approach), Discover (experiential approach), Learn (formal approach). In fact, the Discover environment, stimulates with the contained interviews the linguistic and verbal intelligence, and the naturalistic intelligence by means of the animations. The Learn context involves the logical-mathematical intelligence. By means of the video games, the Play environment makes special use of the visual and spatial intelligence besides fostering prosociality in the multiplayers setting.

The paper is organized as follows. We start by presenting an overview about *gamification*, a process that, besides being more and more recognized as a powerful tool to tackle several challenges in different fields, represents the core and the background of our innovative approach. In the third section, we describe the realization of QPlayLearn, both from a practical and from a conceptual perspective. Finally, we explore one example of how QPlayLearn can be used and report qualitative preliminary results of using the platform in an informative, though informal and funny, event for university students, called Fun in Theory. The present work is aimed at illustrating QPlayLearn idea and how it has been implemented so far. A quantitative assessment of its effectiveness and efficiency must follow, so that we will comment in the concluding remarks on ongoing work that we are conducting along this direction.

## 2. Time to Play: How Gamification Can Support Physics Education

In March 2019, the mobile-game developer Supercell released a video thanking the 100 million people around the world who play its games daily [7]. According to estimates, by the age of 21 the average U.S. citizen has spent more than 10,000 h playing video games [8]—the equivalent of working in a full-time job of 40 h per week for five years. Thus, people have started to wonder if it is possible to channel this enormous amount of human brainpower by designing games with a purpose. For instance, scientists begun to search for video games in which people can solve computationally intractable research problems as a side effect of playing. In fact, despite their capacity to handle vast amounts of data, there are many problems in science that computers cannot yet solve easily or efficiently. It is, therefore, highly desirable to harness innate human abilities to perform tasks that are beyond the grasp of current machines. The challenge is how to turn a research problem into a game, a process known as gamification. Indeed, such a game should not only have a structure that embeds the specific research problem, but also be fun to play.

Despite the idea might sound extravagant or difficult to achieve, there have already been some successful examples, like Foldit [9], EteRNA [10], EyeWire [11], GalaxyZoo [12], and TheBigBellTest [13], which are used to study protein folding, RNA folding, and neuron mapping, explore galaxies near and far, and contribute to the largest Bell test ever performed, respectively. These seminal citizen science initiatives have demonstrated that Games With A Purpose (GWAP) may be a powerful framework to integrate human and machine capabilities to optimize or solve classical and quantum problems otherwise computationally limited, as well as for boosting learning and teaching processes. A special place in this context has been progressively taken by video games, in which diffusion is boosted by the possibility of playing them in the virtual space of internet and by the availability of multi-platform graphic engines [14]. As discussed at length by McGonigal [15], GWAP approaches to research and education are powerful because of the very same game definition and its implications. A game is made by a goal, a set of rules, a feedback system, and voluntary participation. The goal sets the players' purpose, the rules represent in fact a set of opportunities that can be designed to foster creative and strategic thinking, the feedback system reinforces players' motivations, and voluntary participation preserves

players' safety independently of whether they decide to leave or keep up with the game, which in turn ensures an enjoyable game experience and builds up on motivations.

Thus, it is of no surprise that serious games can, as well, be extraordinary tools in pedagogy. In this context, the seminal idea that "playing is children's work" [16] has become the pivot of the widespread and effective method named after Montessori and is considered essential to develop children's emotional, cognitive, motoric, communicative, relational, and creative thinking [17]. In a wider scenario, games can be effective tools to innovate educational, working, and decision-making participatory processes, for a number of reasons that characterize games as compared with reality. In games, the players perform the explicit and voluntary attempt to overcome unnecessary obstacles [18] and can be exceedingly more challenged, while they keep pushing their limits in a competition in its etymological meaning of aiming at something together. Games constructively activate emotions and hopes, while the players' focus remains on activities they are able to perform without fearing failures, possibly in connection and cooperation with other players in a prosocial practice. Games can be epic adventures one would never experience in a lifetime, opening wide spaces and distant times. In games, players are engaged and rewarded by a skill challenge and not lucrative compensations, which succeed in feedback amplification of their motivation: under these circumstances, players' strategic and creative thinking represents a vast amount of ever-lasting and non-material resources for community development, in fact—in its etymon—a form of sustainable economy [15].

In particular, focusing on the context of this paper, quantum mechanics is known as one of the most counter-intuitive and bizarre physical theories, and Niels Bohr's quote, "Anyone who is not shocked by quantum theory has not understood it", is as relevant now as it was when quantum physics was first developed some 100 years ago. Indeed, quantum mechanics involves a change in paradigm perhaps more radical than so many other in the history of human thought. Quantum problems require imagination, diverse thinking, and exploring wide solutions' landscapes. Thus, gamification could represent a precious resource, for both serious games and education. A citizen science approach to quantum physics needs, to start with, the systematic engagement of a huge number of players with diverse creative and strategic thinking. According to McGonigal [15], gamers are an extremely valuable and yet untapped source of participation bandwidth, and whoever figures out how to engage them for real work will get enormous benefit. In fact, the potential of serious games has made them increasingly popular tools in citizen-science approaches to produce innovation in education, decision making, working, and research contexts, and, on the way back, to foster increased competences of citizens in the corresponding areas of knowledge or problem solving.

The idea of exploiting gamification to boost education and quantum science research has been further developed within the innovative ScienceAtHome context [19], animated by a diverse team of data scientists, game developers and designers led by Jacob Sherson, and based at the Physics Department of Aarhus University. The first game, Quantum Moves, was created in 2011, and it has been played 500,000 times by about 10,000 players. It has been used to explore a number of different problems in an integrated experts-citizen scientist research, which includes the optimization of ultracold-atoms experiments [20]. The Turku Group on Quantum Technology (TQT) has used a GWAP approach to measure information flow from/to a qubit as an open quantum system, an especially challenging optimization problem for quantum information theory [21]. The engineering of a Quantum Black Box [22] as a quantum problem solver and optimizers has been proposed within the IQHuMinds initiative (see concept in Figure 1) [23], including the design of input and output human-machine interfaces by means of a cross-disciplinary neuroscience, computer science, and quantum science approach, and the development of education tools aimed at boosting the citizen science resources.

The development of quantum video games in the interactive environment, created by a joined academic and companies effort, was started by TQT in 2016 in the form of Quantum Games Jams [24], noncompetitive hackathons leading to the realization of a

number of video games related to quantum concepts [25]. More recently, it has landed at the 2020 Internet Festival in Pisa [26], accompanied by outreach activities, like the Quantum Circus and Garden, and the proposal of the Quantum Black Box [22]. In this scenario, games realized for quantum computers represent a distinguished experience per se, started by James Wootton at IBM. Rock-Paper and Cat-Box-Scissors; the first multiplayer game, Battleships; TicTacQ; QRogue; and QSnake are selected as non-exhaustive examples [27].

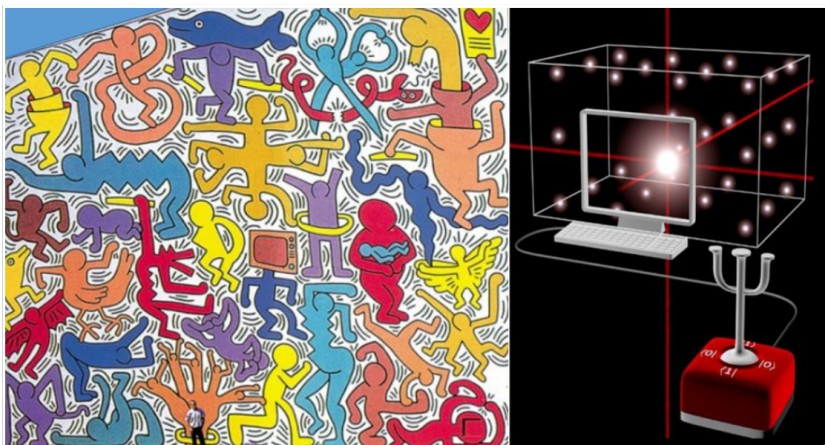

**Figure 1.** IQHuMinds concept of Quantum Black Box. Creativity and intuition from players' crowd, here represented by Keith Haring's artwork "Tuttomondo" in Pisa, are interfaced with the Computer machine via a video game, represented by the psi-joystick, to solve a quantum problem.

Understanding the principles and key conditions for the successful gamification of quantum problems, related to both research and education, is an interdisciplinary endeavour requiring the interaction and collaboration of quantum physicists, game researchers, neuroscientists, and many others. But, because we are on the verge of a new era of quantum technologies, this approach is definitely worth it to be pursued and fostered. In the next section we describe how we convey these ideas in the development of QPlayLearn, that is in order to explicitly address challenges and opportunities of massive quantum literacy.

### 3. QPlayLearn: Quantum Physics for Everyone

Entering the Quantum World means entering a world where we need to abandon our everyday experience of reality. QPlayLearn was born from a group of scientists with a great passion about quantum science, who firmly believe that everyone can learn about quantum physics and its applications. Nevertheless, if for some people the traditional mathematical approach by which quantum science is usually presented can work, for many others a more playful, funny and interactive approach will certainly be more effective. In line with the theory of multiple intelligences, QPlayLearn's holistic perspective stems from the recognition that different types of intelligence dominate the learning process of each individual. Therefore, the platform is conceived to accompany the users in the travel through Alice's rabbit's hole: from the macro- to the microscopic world, offering a number of different paths.

In the section "Play", the games stimulates the interactive participation of the users allowing them to grasp the counter-intuitive features of quantum physics, while they are immersed in a fascinating new world, where the same astonishment we have in front of a science fiction movie or video game is produced. Here, the approach used relies on games and playful experiences, increasing the engagement and the enjoyability of the learning process. In Figure 2 we insert a screenshot of one of the games, "Particle in a box", developed by an interdisciplinary team at Georgia Institute of Technology (Atlanta, GA, USA).

Besides the video games, created by developers from all over the world and carefully selected for QPlayLearn after a long process of research and analysis, the platform contains

also short animations, named Quantum Pills (see Figure 3), explaining fundamental concepts about quantum physics. In order to create them, some of the authors collaborated with VIS (Virtual immersion in science), a spin-off of Scuola Normale Superiore di Pisa, entirely devoted to outreach. Our goal was to create a product having different depths of understanding, which could, therefore, be appealing not only to school kids or the general public but also to quantum physicists, who will find in it a deeper meanings not evident to non-experts. The pedagogical idea is to create engagement in an open space of discussion, where imagination and intuition are guided and supported by a rigorous, non-misleading storytelling of the key concepts, operated with a variety of different languages. This idea connects the primary needs of all users, independently of age and cultural background. Deeper understanding can then built up on top of this engagement, by using progressively more refined symbolic or even formal language, depending on age and formal education degree.

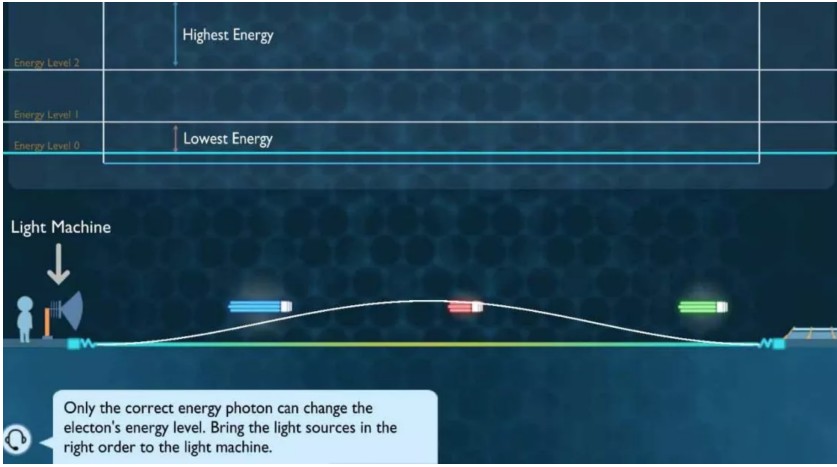

**Figure 2.** Particle in a box is a video game exploring the very first differences between classical physics and quantum physics, developed by an interdisciplinary team at Georgia Institute of Technology (Atlanta, GA, USA).

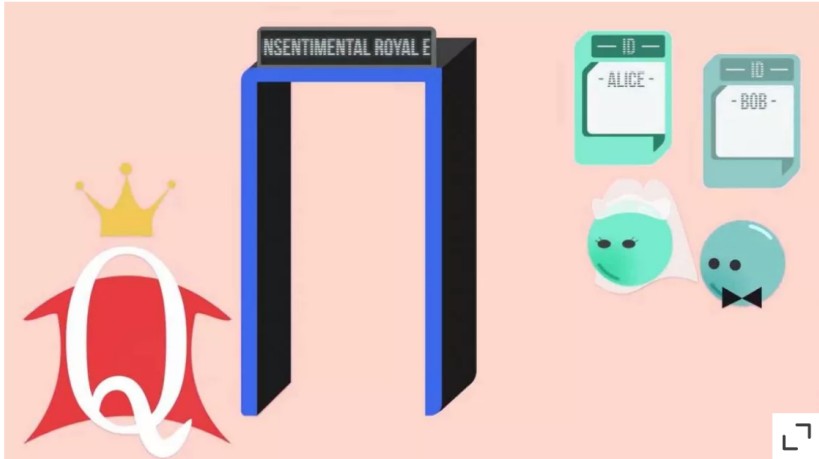

**Figure 3.** In the Quantum Pill about "entanglement" the correlation between two particles is pictured as an (unsentimental) marriage.

The user's journey advances step-by-step, via different and combined learning approaches. Each topic is tackled not only via interactivity and visualization but also via more technical lessons. In the "Discover" part, experts of the field explain with short videos concepts as the Heisenberg principle or the Quantum Measurement exploiting metaphors and deductive examples. Such videos are supported by more thorough texts where the several concepts are presented via scientific experiments, introducing simple

mathematics suitable also for high-school students. A third section, "Learn", enters the core of the quantum theory, and it is instead devoted to a more expert audience, such as university students.

## 4. Results and Discussion: The Fun in Theory Experience

Overall, our method is based on a 3-fold approach: (1) building intuition and engagement through games and videos; (2) understanding physical concepts through easy-to-follow and scientifically accurate descriptions, graphics, animations and experiments; (3) acquiring formal understanding through the mathematics. The idea is that each user can begin from the approach that feels easier or closer to them, and then—if they like—explore the others. Eventually, it is the combination of the different modalities that allows the understanding of quantum physics to shift and expand. In fact, QPlayLearn offers to the users the possibility of choosing the best suited learning approach and to educators the opportunity to choose how to use the platform.

In the following, we explore the utilization of the section "Play" in the context of Fun in Theory event, aimed to inform the bachelor's and master's mathematics and physics students in the University of Turku about the Theoretical Physics courses, master's program and line of research. The Fun in Theory is an event created and organized by the laboratories of Theoretical Physics and Quantum Optics in the University of Turku. The event is held annually, and for the University's students has now become a tradition. The event is intended to be socially interactive and informative, and casual and funny at the same time. For the participants, there are no requirements of previous knowledge in theoretical physics.

Fun in Theory brings together students and staff members, with the overall aim to inform about the theoretical physics courses and study possibilities in the University of Turku. Importantly, it aims at introducing students with different backgrounds to the world of theoretical physics in an understandable and playful way.

The event is divided in two parts: a 40-min general lecture conducted via a presentation—the *theory* part, and a 2-h game, in which students participate in groups and compete for small prizes—the *fun* part. During the lecture, students learn in general terms what theoretical physics researches, what impact it has on our society, why it is important to study and develop different branches of theoretical physics, and how it can be done at the University of Turku. The game consists always of small tasks and is related to the subjects in theoretical physics.

Going back to QPlayLearn, the section "Play" adapts very well for the "fun" part of the event, for several reasons: it predisposes to social interaction inside the groups, amplifying at the same time the competition between the groups, and creates a playful environment, while giving an insight on quantum physics basics in general terms. Moreover, quantum physics courses are of the greatest importance in bachelor's and master's programs, and certainly quantum science constitutes an important part of nowadays theoretical physics research. In this context, QPlayLearn "Play" becomes a great tool to show the students a glimpse behind the curtain of what quantum mechanics courses contain. Combined with the initial lecture about the importance and societal impact of theoretical physics, and specifically also of quantum physics, it contributes to provide a bigger picture on this topics to younger students.

This year Fun in Theory was held as a virtual event (due to Covid outbreak) in Discord, an instant messaging and digital distribution platform, and the game part consisted of four educational computer games related to quantum mechanics, quantum optics and special relativity. As the games related to quantum mechanics, we used two games from QPlayLearn-Play section, named *Psi & Delta* [28] and *Quantum Playground* [29]. A game revealing secrets of quantum optics laboratory, precisely a quantum optics laboratory playful simulation, was *Quantum Game with Photons 2* [30]. For special relativity, we used a game called *Velocity Raptor* [31]. After each game, the students filled a questionnaire, which we used as a basis to decide the winner team of the whole game. The game part of Fun

in Theory gave us a possibility to collect a first qualitative feedback about the potential of QPlayLearn resources and setup. Because of the 2-h limitation, we associated only to the game Psi & Delta a larger questionnaire for a preliminary evaluation of the enjoyability and effectiveness of the game and of the students' potential interest in learning theoretical physics also by means of educational games.

The participants were divided into 3 groups of 5 people and one group of 4. Because all the participants had different competences in theoretical physics, in each group, there was at least one person who had passed through all quantum mechanics courses for theoretical physics bachelor's students.

The order of the games, the duration of the play time, and the estimated time for filling the questionnaire after each game were as follows:

- Psi & Delta, 40 min, 15 min;
- Quantum Playground, 10 min, 5 min;
- Quantum Game with Photons 2, 15 min, 5 min;
- Velocity Raptor, 15 min, 5 min;
- Extra time for completing everything, 10 min.

Each team had its own vocal and text channel in Discord, so the team members could interact with each other. The teams were instructed to play so that one team member shared a screen with the game with other members or that everyone would play by themselves, or to combine the two methods. The participants were encouraged during the play time to freely discuss in their teams everything that happened in the game, and also everything related to physics concepts and phenomena presented in the game. After each game, participants filled in the questionnaire individually, without any discussions with each other.

In particular, after the game Psi & Delta, 18 Fun in Theory participants filled in the questionnaire related to the game. A quantitative assessment of this educational setting is beyond the scope of the present paper. Nonetheless, we report here below some information about the questions and related answers as an indication for further development and considerations for a suitable evaluation. The majority of the participants was composed by Physics and Astronomy students. Among the students, there were only 7/18 who never studied theoretical physics on any level, whereas only 4/18 already attended and passed the course of quantum mechanics basics. In the questionnaire, we used three types of questions: (1) questions with short free answer, (2) questions with given answer variants to choose from, and (3) questions with rated answers, scaled from 1 to 5. Overall, we had a positive response and a great engagement during the event. The students agreed in evaluating the game as a powerful tool for integrating traditional learning: none of them would prefer to have only lectures, even if all of them claimed the necessity of further explanation for deepening the concepts. All the students claimed to get an intuition of what might be taught in the course of quantum mechanics, after playing the game, and they all agreed that it was effective for presenting basics of quantum physics. In general, they all would like to have games in the theoretical physics courses to teach some concepts, assuming that games are created especially for this purpose.

## 5. Conclusions

QPlayLearn is a platform for wide and diverse use, intended to tailor education processes on quantum science on the many talents of different users. We have discussed how this goal is pursued by designing around each quest different types of activities, i.e., "Play", "Discover", and "Learn", which imply the use of diverse types of thinking and relationship skills with the environment. Because of this flexibility, QPlayLearn can be adopted by educators as largely as one can imagine at first.

The platform is completely free—the contents can be used with Creative Commons license—and aimed for a broad audience. While the method is general, we aim at tailoring our contents to certain targets, and we are specifically developing workshops and courses for high schools, companies, and universities. This is an essential trait, since we eventually

aim at operate a diffuse and massive cultural change, building up literacy and awareness about how quantum physics works.

In fact, we propose to foster QPlayLearn's approach in developing tools and strategies for quantum physics education and learning processes, independently of age and previous instruction context. To this aim, we wish to investigate the potential of the approach, how to fully exploit its opportunities and the manner with which its possible limitations can be transformed into innovations for learning and teaching processes.

As a first step along this program, we focused ourselves on among the possible uses of the QPlayLearn platform, that relying on the "Play" approach, which was ideal for the Fun in Theory game activity and for achieving the goal of the event. Though qualitative, the feedback from the Fun in Theory activity suggests that video games work to motivate, light curiosity on, and cooperate to effective and efficient learning process. While the aim of the present work is to illustrate the QPlayLearn idea, a number of questions arise about the effective resolution of the deeper conceptual issues and achievement of the expected learning outcomes. To this aim, a systematic, dedicated physics education research study is required. In fact, learning from the experiences with the survey about the QPlayLearn game Psi & Delta, we have started a collaboration with the physics education research group of professor Marisa Michelini (University of Udine, Udine, Italy), aimed at designing and validating suited measurement tools to quantitatively investigate effectiveness and efficiency of the QPLayLearn approach in high schools. The results are aimed at iterative improvements of the platform and its corresponding education environment.

The present QPlayLearn setting opens up three future perspectives. One is towards enhancing the connection between creative thinking and practical application of concepts. This will be performed by completing the quest environments "Play", "Discover", "Learn" with two new learning environments: "Apply" and "Imagine". The first new section will be devoted to practice: the users will be challenged to run code samples on real world quantum devices. For each entry of the dictionary, there will be tutorials and small code projects so that the users can apply immediately what they learn in the other sections of QPlayLearn. In order to develop the "Apply" part, we are collaborating with Strangeworks, a platform that enables its users to experience quantum computing and exploit it for any problem and research in a simple way. As for the "Imagine" section, we envision an environment where the learning process be totally free from any restriction and everyone can express their grandeur with respect to the quantum world through the artistic language she/he prefers. This will be the area devoted to creativity at its most, for science-fiction storytelling and art. Our measurement tools will have to be tailored correspondingly, to be able to assess how each type of activity impacts on the overall learning process. Second, accessibility is also linked to verbal language literacy, and we think that a multiple language version of the QPlayLearn platform should be in order. A third and most significant direction is about extending the class of beneficiaries to 0–99 years old users, by means of a suited pedagogical strategy and tools. This is, we believe, an authentic priority, that we consider in fact an unprecedented challenge.

**Author Contributions:** All authors contributed to all aspects of the research. All authors have read and agreed to the published version of the manuscript.

**Funding:** The work has been partially funded by IBM Finland, Algorithmiq Oy, and the Academy of Finland via the Centre of Excellence program (Project no. 336814).

**Institutional Review Board Statement:** Not applicable.

**Data Availability Statement:** No datasets were generated or analysed during the current study.

**Acknowledgments:** We thank IBM Finland and Algorithmiq Oy for partly funding this project. S.M. acknowledges financial support from the Academy of Finland via the Centre of Excellence program (Project no. 336814).

**Conflicts of Interest:** The authors declare no conflict of interest. The funders had no role in the design of the study; in the collection, analyses, or interpretation of data; in the writing of the manuscript, or in the decision to publish the results.

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
