# Peer review of "Quantum Physics Literacy Aimed at K12 and the General Public†"

_universe, doi:10.3390/universe7040086_

Round 1

Reviewer 1 Report

Referee report on “Quantum physics literacy aimed at K12 and the general public”

First of all: are the authors serious that quantum literacy should be reached for already in Kindergarten?

Scientific literacy is a long standing goal of physics education and this paper introduces concepts to reach for a specific “quantum literacy”. The premise of their argument (according to the abstract) is that “classical” physics can utilize everyday examples to foster the understanding. All physics teachers (me being one of them) would probably agree that this is in large part just wishful thinking. Indeed, physics is considered by itself a highly abstract enterprise and the literature about “alternative frameworks” about “classical physics” is vast. I suggest that this part should be modified accordingly.  Be that as it may, any attempt to improve scientific literacy (of which any alleged “quantum literacy” is a subset) should certainly be welcome.

Furthermore the authors claim that a better understanding of this “new way of looking at the world” is essential for general welfare and “smart community building”. While quantum technologies are increasingly affecting our lives, I am not so sure if this is the right way to look at it. The following relates to my point above: Although also pre-quantum physics is only so poorly understood by many, most people are still enjoying their lives in a world which is dominated by technology already (although not so much of the quantum 2.0 type yet). But still and again: any effort is welcome.  I suggest a rewording to avoid exaggerated expectations.

Next, the authors relate their effort to Garden’s concept of “multiple intelligences”. I must confess that I am not familiar with this concept; however, the brief exposition indicates the importance of different channels or representations for individual types of learners. These seem fairly standard assumptions to make about the learning process in general. What it comes to is the issue of “gamification”. Here the authors make a compelling case for resources that can be used “with a purpose” so to say. However, the connection with “citizen science” is two-edged, since here quite often the participants do not grasp the scientific background properly (and do not have to). And (again): The opportunities to foster learning by gamification are certainly not restricted to quantum physics. In fact, they are not restricted to physics at all….

Let me turn to the “Results and discussion”. I certainly read the paper alongside playing on the corresponding website, watching some of the videos and reading the texts. Some of the following remarks are therefore not exactly part of my paper-referee job. In general I dislike the dramatically elevated tone and bombastic language (this relates to both, paper and website). Yes, QM is counterintuitive and all. But in an educational context the foremost goal should be to demystify quantum physics since nobody tries to understand something which is allegedly unintelligible. This is certainly related to the issue what one really takes to be the essence of quantum physics. In my feeling the current project puts too much emphasize still on outdated notions like “complementarity” and “wave-particle duality”. True, the website “expert-text” on wave-particle duality explains nicely that this expression is rather a metaphor (a rare example in the educational literature!). But the best way to illustrate this is to move to at least a 2-particle system. I believe that it is almost a tragedy that in teaching QM low dimensional examples are discussed as nauseam. Here, the difference between configuration space and ordinary 3-space plays no role and the confusion that psi is some actual wave is promoted. In any event, to me the key feature of QM is that it does not provide a space-time picture of the events between preparation and measurement. Some of the games may support this misunderstanding (others do not – for sure). The quantum tic-tac-toe, for example, is brilliant.

The evaluation part at the end is certainly suffering from a too small statistics. Perhaps it should be shortened to extract just some rough tendencies.

In summary, a very enjoyable read and an exciting project. However, unfortunately there are only very few references to the physics education literature given. Perhaps also some of Antje Kohnle’s work could be relevant. These should be filled in. If (as explained above) the wording could be modified to avoid exaggerated expectations and too much bombast I support its publication completely.

Cheers,

Reviewer 2 Report

The article presents a series of small computer games meant to demonstrate or illustrate some aspects of quantum physics to the general public, and discusses the responses of University students to questions on their experience with playing these games. The computer games themselves seem to be of the typ which is most often seen in science festivals and exhibits, and this is acknowledged in the paper. The paper is an interesting read and touches very important subjects, although I must say it has very little in common with the research methods of physics education. There is a good introduction about gamification, but actually there is no precise identification of what concepts of quantum physics are meant to be gamified/understood by students or general public within each game. By a 'precise identification' I mean not just 'this game is about energy levels' but for example, what concepts about energy levels are included in the game? Has the research on student misconceptions of energy levels been evalutated? In which sense the game is meant to provide a non misleading picture of energy levels? What are the learning outcomes? And so on. Also, questions about interest/engagement of students are answered only at the level of qualitative indications in the above mentioned questionnaire, without delving with any one of the numerous theoretical paradigms which have been constructed in science education for studying interest/engagement/involvement/etc.

Of course, answering or even attempting an answer to the above questions would require a completely different article. As far I can judge, the article is a report on a very interesting and probably promising outreach activity, with no special focus neither on learning outcomes, or results in terms of the participants' engagement. The article is written with care and a captivating style and, as I wrote above, interesting to read. If this kind of articles is within the scopes of the journals, which I am unable to judge, I recommend accepting it with no modifications.

Reviewer 3 Report

The authors report on an online platform they have developed (QPlayLearn), that provides multilevel education on quantum mechanics using games, animations and easy-to-understand explanatory videos by experts. More features and learning environments are foreseen for the future.

 General Comments

  • There is not enough discussion about how the QPlayLearn learning environments/activities are connected to Gardner's Theory and to each of the intelligences that are listed in lines 52-53. 
  • QPlayLearn as the main subject of the paper, should be described more thoroughly, maybe in a different section before the “Results and Discussion” section. The authors could add here also the connection to the multiple intelligences theory (see previous comment)
  • A more essential title is needed for section “Methods”.
  • There are not any references in the manuscript for all Figures presented.

Specific Comments

  • Line 36. AI abbreviation not defined
  • Line 125: ScienceAtHome needs reference
  • Line 132: Quantum Black Box reference is given later. It should be moved here
  • Line 137: Quantum Games Jam needs reference
  • Line 138: Unknown word “hackatons”. Maybe hackathons?
  • Line 169: srl abbreviation not defined
  • Line 216: Discord. A footnote is needed about that
  • Line 219-221: Some titles appear in italics, while other titles normally
  • Line237: The previous list shows the time for filling the questionnaires. Here it is said that 10 more minutes were given extra? If so, why isn’t it included in the previous list?
  • Line263: Sentence does not make sense.
